# Free-standing supramolecular hydrogel objects by reaction-diffusion

Matija Lovrak[1], Wouter E.J. Hendriksen[1], Chandan Maity[1], Serhii Mytnyk[1], Volkert van Steijn[1], Rienk Eelkema[1] & Jan H. van Esch[1]

Self-assembly provides access to a variety of molecular materials, yet spatial control over structure formation remains difficult to achieve. Here we show how reaction–diffusion (RD) can be coupled to a molecular self-assembly process to generate macroscopic free-standing objects with control over shape, size, and functionality. In RD, two or more reactants diffuse from different positions to give rise to spatially defined structures on reaction. We demonstrate that RD can be used to locally control formation and self-assembly of hydrazone molecular gelators from their non-assembling precursors, leading to soft, free-standing hydrogel objects with sizes ranging from several hundred micrometres up to centimeters. Different chemical functionalities and gradients can easily be integrated in the hydrogel objects by using different reactants. Our methodology, together with the vast range of organic reactions and self-assembling building blocks, provides a general approach towards the programmed fabrication of soft microscale objects with controlled functionality and shape.

[1] Department of Chemical Engineering, Delft University of Technology, van der Maasweg 9, 2629 HZ Delft, The Netherlands. Correspondence and requests for materials should be addressed to R.E. (email: r.eelkema@tudelft.nl) or to J.H.v.E. (email: j.h.vanesch@tudelft.nl).

Over the past decades the self-assembly (SA) of a wide variety of building blocks has become an established technology for the bottom–up fabrication of objects and materials with structural features ranging from nano- up to micrometre length scales[1,2]. The general approach to create objects and structures of dimensions beyond the nanoscale is to increase the size of the building blocks. However, this comes with new challenges – to fabricate these larger building blocks with sub-micron features and to manage the delicate balance of forces between building blocks, diffusion, and inertia. Reaction–diffusion (RD) plays a key role in natural structure-forming processes, including SA and cell proliferation, which control the formation of a wide variety of structures, ranging from actin filaments, extracellular matrix, to organs and skin patterns[1,3–5]. In RD, two or more reactants diffuse when locally released at different positions, giving rise to spatial concentration patterns, which may lead to local structure formation, like Liesegang rings, polymerization or SA, on reaction[6,7]. In recent years, almost exclusively inorganic RD systems have expanded into a wide range of scientific and technological areas, such as biomineralization[8], microfabrication[9–13], the formation of microlenses[7,9,14], the formation of microparticles and microspheres[15,16] and dynamic materials[17]. The reported RD patterns and structures reach high levels of complexity and resolution[18], but so far the application of RD to control structure formation of organic materials has been limited. Organic chemistry offers both an extremely wide range of chemical reactions and functional materials, as well as the possibility to precisely control chemical kinetics across multiple time scales. Only a few examples of RD using biological reactants have shown that chemical gradients can be programmed using DNA-based autocatalytic reaction networks[19–21], and enzymes[22,23]. With organic compounds, RD has been used to fabricate anisotropic structures[24], and, only very recently, to achieve spatial control over the formation of micro-objects by a polymerization reaction[25].

Here we describe the combined RD and SA of a supramolecular gelator leading to the formation of free-standing macroscopic structures with controllable shape, size, and chemical functionalization. In this system, multiple components diffuse towards each other, to react at the crossing of the diffusional fronts and form hydrogelator molecules, eventually leading to a supramolecular structure through SA. We show that the patterned structures can be chemically functionalized and functionalization can be used to form permanent chemical gradients inside the final structures. The methodology developed here provides a general approach towards the programmed fabrication of soft microscale objects with controlled functionality and shape, and we anticipate that it may be applied for the creation of new functional soft biomaterials with a wealth of shapes, sizes, and chemical functionality.

## Results

**Fabrication of 1D reaction–diffusion pattern**. We use RD-SA to control the spatial distribution of supramolecular materials (Fig. 1). RD-SA requires a multicomponent reaction inducing a SA process, to allow independent diffusion of reactants and SA of products at the crossing of the diffusional fronts. We have recently reported a supramolecular hydrogelator that is formed in the gelation medium by a multicomponent chemical reaction[26]. In this work, the acid-catalysed reaction of hydrazide **H** and aldehyde **A** leads to the formation of the trishydrazone gelator $HA_3$ (Fig. 1a)[27]. When $HA_3$ reaches concentrations above its critical aggregation concentration (CAC), it self-assembles into fibres and *in situ* forms a non-transparent supramolecular network. The rate of formation of the supramolecular network

is controlled through the hydrazine-forming reaction, which depends on the concentration of reactants and the presence of an acid catalyst[27,28]. In our RD-SA approach, reactants **H** and **A** diffuse over a distance and react on crossing of the moving fronts, forming $HA_3$, which subsequently self-assembles into a supramolecular material. In a typical RD-SA experiment, **H** and **A** are placed at the opposite sides of an agar gel matrix (acting as a diffusion medium), and left to diffuse and react over time (Fig. 1b). We observed that the formed $HA_3$ forms a non-transparent supramolecular structure within the agar matrix, manifesting as the appearance of a turbid line structure at the intercept of the two diffusion fronts. Structure formation started after ~7 h, and the formed line typically achieved a width of 2.5 mm within 24 h (Fig. 2a and Supplementary Movie 1) when using 2 cm wide agar gels at general experimental conditions. The supramolecular structure forming the line (referred to as a 1D pattern) extends vertically from the bottom to the top of the 3.5 mm-thick agar matrix, and is stable for months.

**Fabrication of more complex RD patterns**. With this result in hand, we set out to explore the potential of RD-SA to control shape and dimensions of the formed structures. We started by investigating the influence of initial localization of reactants, diffusion distance, and type of reactant on the resulting shapes. By positioning the reactants in pre-designed locations, we anticipated obtaining a variety of two-dimensional (2D) shapes. We observed local formation of the supramolecular structure at the crossing of diffusional fronts of **H** and **A** (Fig. 2a). More complex shapes and patterns such as waves, grids, circles, and triangles were easily made using three types of approaches for reactant injection: by cutting reservoirs for the reactants into the agar matrix (Fig. 2a,b,f and Supplementary Movie 2), by placing reservoirs into a polydimethylsiloxane (PDMS) mould placed under the agar matrix (Fig. 2c–e), or by printing droplets with reactants as point sources on a flat agar matrix (Fig. 2g). When the PDMS mould consists of a grid of reservoirs, the formed structures consist of several connected segments generated in the regions where the reactants, after diffusing from their reservoir, meet and react (Fig. 2e). A close look at a segment reveals that the centre is more opaque than the edges as a consequence of the neighbouring reservoirs being physically further apart diagonally than side-by-side. This separation, in turn, causes the diffusional fronts of **H** and **A** to overlap with time delay at the edges. To demonstrate that the choice of reactants is important for the final appearance of produced patterns, we prepared a pattern using structurally similar 3,4-dihydroxybenzaldehyde (**A***) simultaneously with **A** (Fig. 2f). It can be seen that the $HA_3^*$ pattern is thicker and looks precipitate-like compared to the $HA_3$ pattern. By precise positioning of reactant reservoirs with the aforementioned methods, we were able to generate a range of shapes, including squares, circles, grids, triangles, and even letters (Fig. 2g).

**Mechanical properties and morphology of hybrid gel network**. We then aimed at making free-standing objects using RD-SA. Making free-standing objects requires removal of the diffusional matrix after completion of the RD-SA process. Also, the objects have to be sufficiently mechanically strong to carry their own weight. To facilitate straightforward removal of the matrix, we used calcium alginate instead of agar as a diffusional matrix. Alginate gels can be dissolved and subsequently removed by the addition of ethylenediaminetetraacetic acid (EDTA) solution, removing the cross-linking divalent ions between the alginate chains. We confirmed that changing the matrix from agar to

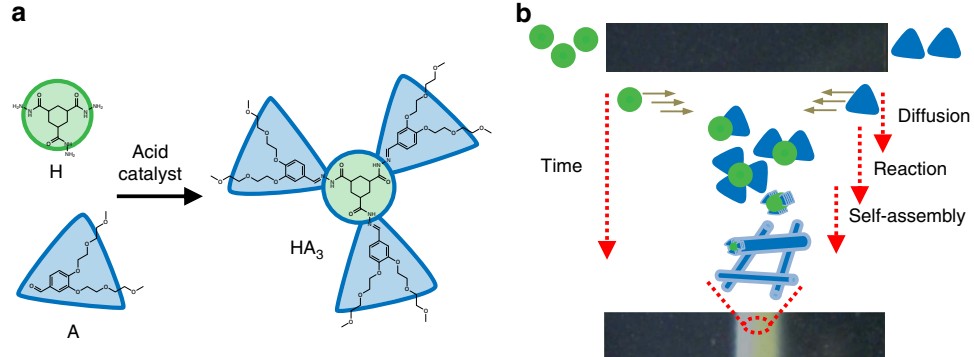

**Figure 1 | RD of a two component self-assembling gelator.** (**a**) Hydrazide **H** and aldehyde **A** react to form gelator **HA₃** under ambient conditions in water, with acid acting as a catalyst. (**b**) The space-time plot illustrates the RD-SA process. Reactants **H** (green circle, left) and **A** (blue triangle, right) are placed on the opposite sides of the agar gel matrix. Over time **H** and **A** diffuse through the matrix, and react on crossing to form gelator **HA₃**, which self-assembles into a fibrous gel network. This process results in the formation of a turbid white line consisting of gel fibres, within the diffusion matrix.

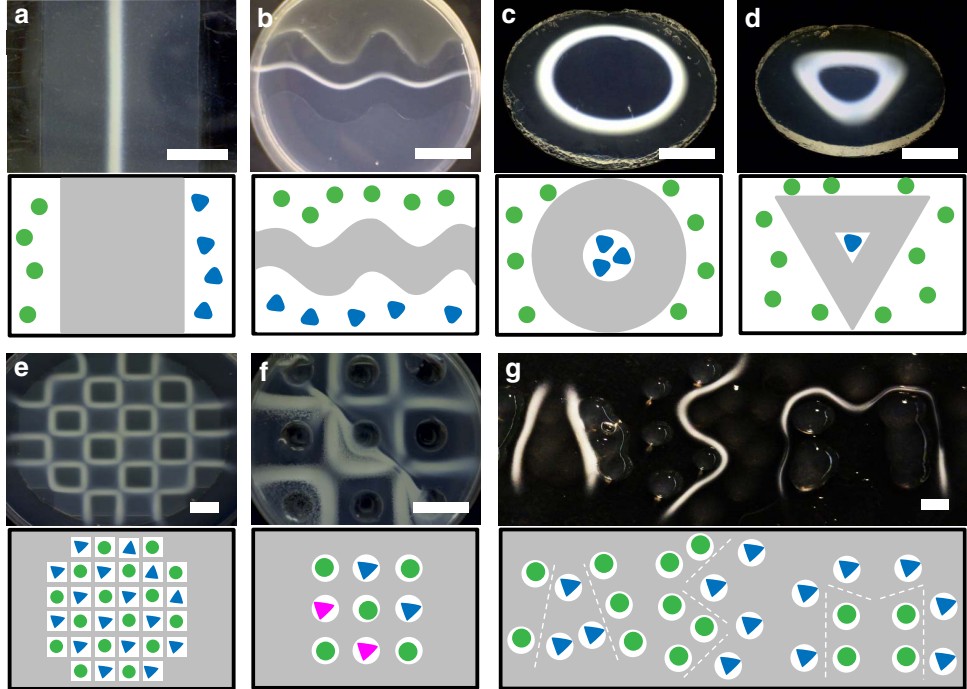

**Figure 2 | The initial location of reactants controls structure formation in RD-SA.** In **a**–**g**, the gel structure formed through RD-SA (photograph, top) resulted from a starting point configuration as shown in the cartoon below. In the cartoons, the grey areas denote the agar matrix, white areas are the reservoirs, which are filled with hydrazide **H** (green circles) and aldehyde **A** (blue triangles). (**a**) A 1D pattern formed by RD of **H**, from the left, and **A**, from the right, in an agar diffusion matrix. Method: reservoirs in the agar matrix. (**b**) By shaping the agar matrix, the RD-SA formed patterns were obtained in different shapes, such as waves. Method: reservoirs in the agar matrix. (**c,d**) To show the versatility of the RD-formed gel structures, different designs were used to obtain a circle and a triangle. Method: a PDMS mould below the agar matrix. (**e**) A grid formed by RD-SA. Method: a PDMS mould below the agar matrix. (**f**) An RD-SA grid made from aldehydes **A** and **A\*** to compare the difference in SA behaviour, (magenta: **A\***, blue: **A**). Method: reservoirs in the agar matrix. (**g**) Using the RD-SA approach to 'write' our research group name (ASM). Method: droplets of reactants in agar solutions were placed on top of an agar matrix, to diffuse and form supramolecular structures at the intersection of the diffusion gradients (Supplementary Methods). Scale bars: 1 cm.

alginate does not significantly influence the resulting RD-SA pattern (Supplementary Fig. 11).

We investigated the mechanical properties of the RD-SA structures in alginate gels. Alginate/**HA₃** hybrid gels with varying compositions were subjected to compression tests (see Supplementary Fig. 3 and Supplementary Methods for details of preparation). We find that the formation of a supramolecular network inside an alginate gel leads to a hybrid material displaying a hugely increased yield stress when compared to pure gels formed from either alginate or **HA₃** (Fig. 3a,b). For instance, a material consisting of 1.5% alginate gel and a hydrazone network made from 40 mM **H** (denoting the initial concentration of **H** in alginate, with **A** in excess) has a yield stress of 62 kPa, compared to only 7 kPa for the pure 1.5% alginate gel, and 0.7 kPa for a pure 30 mM **HA₃** gel (see Supplementary Methods for detailed explanation).

To explain why the mechanical properties of alginate/**HA₃** improved compared to the separate gel networks, we investigated

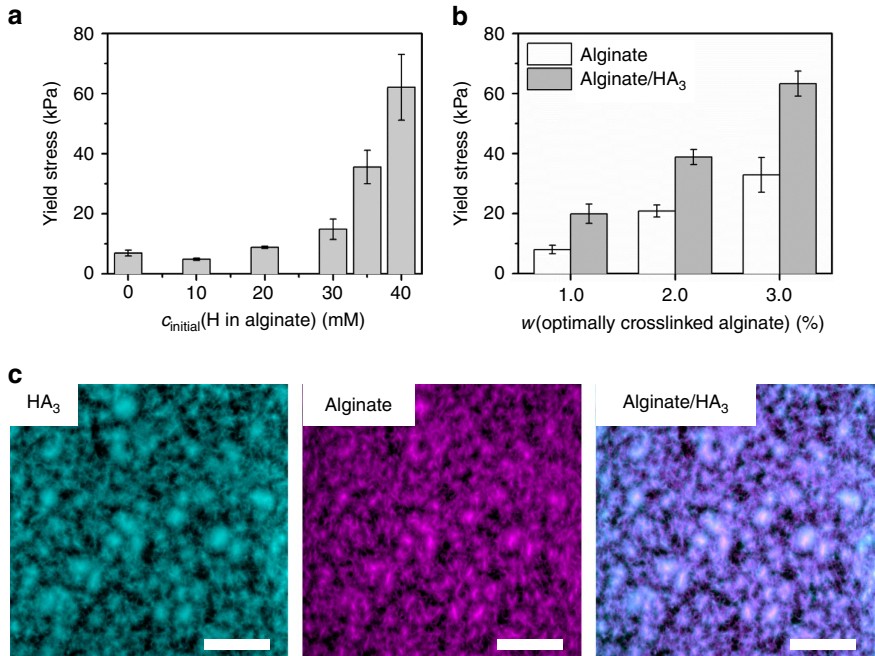

**Figure 3 | Mechanical properties and structure of the formed hybrid network gels. (a)** The effect of the initial concentration of hydrazide **H** in alginate on the compressive strength of the alginate/**HA₃** hybrid network material. Compression rate was 50 mN min$^{-1}$. **(b)** The effect of the concentration of the optimally cross-linked alginate on the compressive strength of alginate and alginate/**HA₃** hybrid network material. Compression rate was 250 mN min$^{-1}$. **(c)** Confocal fluorescence micrographs of **HA₃**, alginate and overlay images of alginate/**HA₃**. **HA₃** is labelled with a pyrene-functionalized benzaldehyde (**AP**, $\lambda_{ex} = 405$ nm) and alginate is labelled with BODIPY TR ($\lambda_{ex} = 543$ nm). The error bars in **a** and **b** were calculated as a s.d. of at least three measurements. Scale bars: 50 μm.

the microstructure of the formed gel patterns with confocal fluorescence microscopy. To distinguish between the self-assembled hydrazone structure and the alginate matrix, we labelled **HA₃** with a pyrene fluorophore **AP** (see Supplementary Fig. 1 and Supplementary Methods), and alginate with a BODIPY TR fluorophore (Supplementary Methods). As can be seen in Fig. 3c, the fibres of **HA₃** are co-localized with the alginate chains, which suggests that **HA₃** and alginate form a hybrid network material. It is known that networks consisting of two different gels often exhibit vastly improved mechanical properties[29]. We did not investigate how the interactions or synergy between **HA₃** and alginate lead to improved mechanical properties, but we hypothesize that **HA₃** fibres wrap around the alginate chains and create cross-links between the alginate chains in addition to already existing calcium cross-links, most likely in a similar fashion as recently has been reported by Kiriya *et al.*[30] This additional cross-linking, in turn, would improve the mechanical properties of alginate/**HA₃** hybrid network material.

**Fabrication of free-standing objects.** The excellent mechanical strength of the alginate/**HA₃** hybrid network material encouraged us to further investigate the possibilities of making free-standing objects. We used a similar approach as in the experiments with agar, now including an additional step for removing the alginate matrix (Fig. 4a). Following the proposed scheme, we successfully made free-standing objects with distinct shapes (Fig. 4b), where the width of the lines constructing the object is in the millimetre range and the size of the full object is in the centimetre range. We subsequently looked at downscaling the RD-SA process to extend RD-SA to applications at sub-milli-metre length scales. Creating diffusion patterns through the manual cutting approach did not allow us to achieve the required

resolution. Therefore, we turned to wet stamping (WETS) as an alternative approach[7,31].

In the WETS approach, a substrate and a stamp were both made from alginate or agar gel. The substrate contained **A** and the stamp contained **H**. On bringing the stamp and the substrate into contact, **H** from the stamp diffuses into the substrate and **A** from the substrate diffuses into the stamp. After 60 min of stamp-substrate contact, the stamp was removed and the substrate was left standing overnight to allow RD-SA to take place. We observed formation of **HA₃** patterns in the substrate. The patterns were examined using a confocal microscope before and after dissolution of the alginate substrate (Fig. 4d). Objects as small as 300 μm (measured as the width of a single line) were successfully prepared (Fig. 4c). When we tried to use the stamp with a 200 μm feature size with 200 μm spacing, objects could not be successfully separated from each other on dissolution of the alginate substrate.

**Functionalization of patterns and free-standing objects.** With all the tools established, we set out to explore the potential of RD-SA to control chemical differentiation and functionalization of the formed structures. Exploratory experiments were performed in agar diffusion matrices. We showed previously that the hydrazide-aldehyde reactive gelator system is very tolerant towards the use of different aldehydes[32]. Combined with RD-SA, this feature can be exploited to create patterned supramolecular gels with spatially differentiated zones of chemical functionalities, by placing different aldehydes at different locations before diffusion. Indeed, by using different fluorescent aldehydes at different diffusion locations with this approach, we were able to fabricate 2D gel patterns with stable spatially varying differences in fluorescence and colour that were visible by confocal fluorescence microscopy and even by eye (Fig. 5a,b).

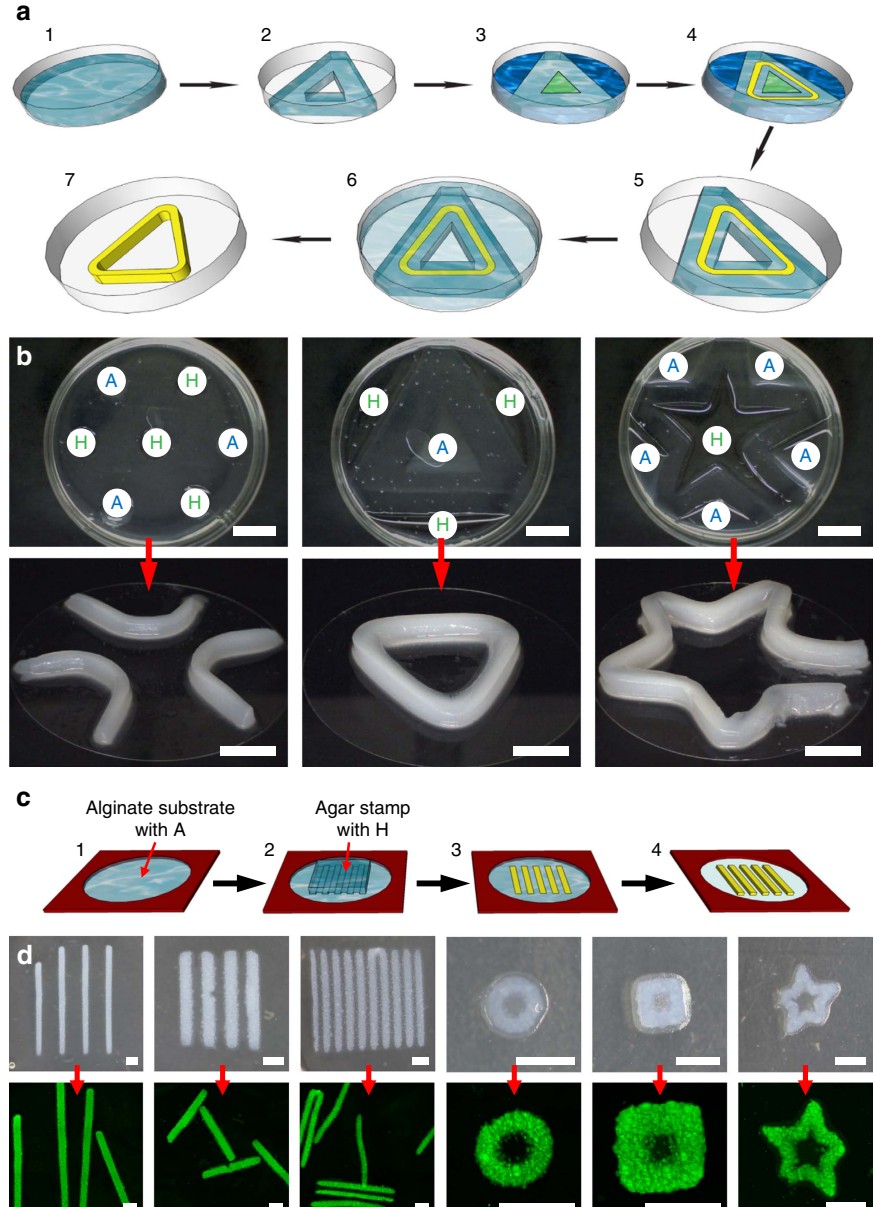

**Figure 4 | Free-standing hydrogel objects.** (**a**) General method of preparing free-standing objects using the cutting approach. (1) An alginate hydrogel is prepared in a Petri dish. (2) An arbitrary shape is cut out of the alginate. (3) Solutions of hydrazide **H** (green) and aldehyde **A** (blue) are placed into the reservoirs. (4) **H** and **A** diffuse through the alginate matrix and react at the diffusion fronts to form **HA₃**, which then self-assembles into a gel structure (yellow). (5) The remaining solutions are removed. (6) A solution of EDTA is poured into the Petri dish until it completely covers the alginate containing the formed pattern. (7) After all alginate is dissolved (as observed by visual inspection), the remaining solution is removed and the free-standing hydrogel object is obtained. (**b**) Free-standing hydrogel objects prepared using the cutting approach. (**c**) A WETS approach for the preparation of micro-sized free-standing objects: (1) A 1 mm-thick layer of alginate is prepared on a glass slide and loaded with **A**. (2) A stamp containing **H** and rhodamine B-benzaldehyde (**AR**) is placed on the substrate for 60 min and is then removed. (3) After standing overnight, the pattern of **HA₃** appears. (4) Dissolving the remaining substrate produces the free-standing objects. (**d**) Photographs and confocal images of the patterns of **HA₃** and free-standing objects. The dimensions of the stamps for lines (from left to right): 500 μm feature/1500 μm spacing, 500 μm feature/500 μm spacing, and 300 μm feature/300 μm spacing. Scale bars: 1 cm in **b**; 1 mm in **d**.

Next, we investigated the possible formation of permanent chemical gradients within the formed supramolecular patterns. Such chemical gradients can, for instance, be useful to control cell differentiation in space[33]. We mixed **H** in agar. **A** was mixed with an aldehyde-functionalized fluorescent probe in buffer and allowed to diffuse in from one side of the agar matrix. After 2 weeks we analysed the material within the agar matrix using confocal microscopy, showing an 8 mm wide permanent fluorescence gradient within the formed fibre network (Fig. 5c,d). The experimental time was significantly longer than in the formation of 1D patterns, because here the diffusion of **A** was hindered by immediate formation of **HA₃** at the gel/solution interface (see Supplementary Fig. 3 for the experimental details). This gradient in fibre network density was quantified by measuring the fluorescence intensity over the full distance. Surprisingly, the gradient was not completely gradual, but showed

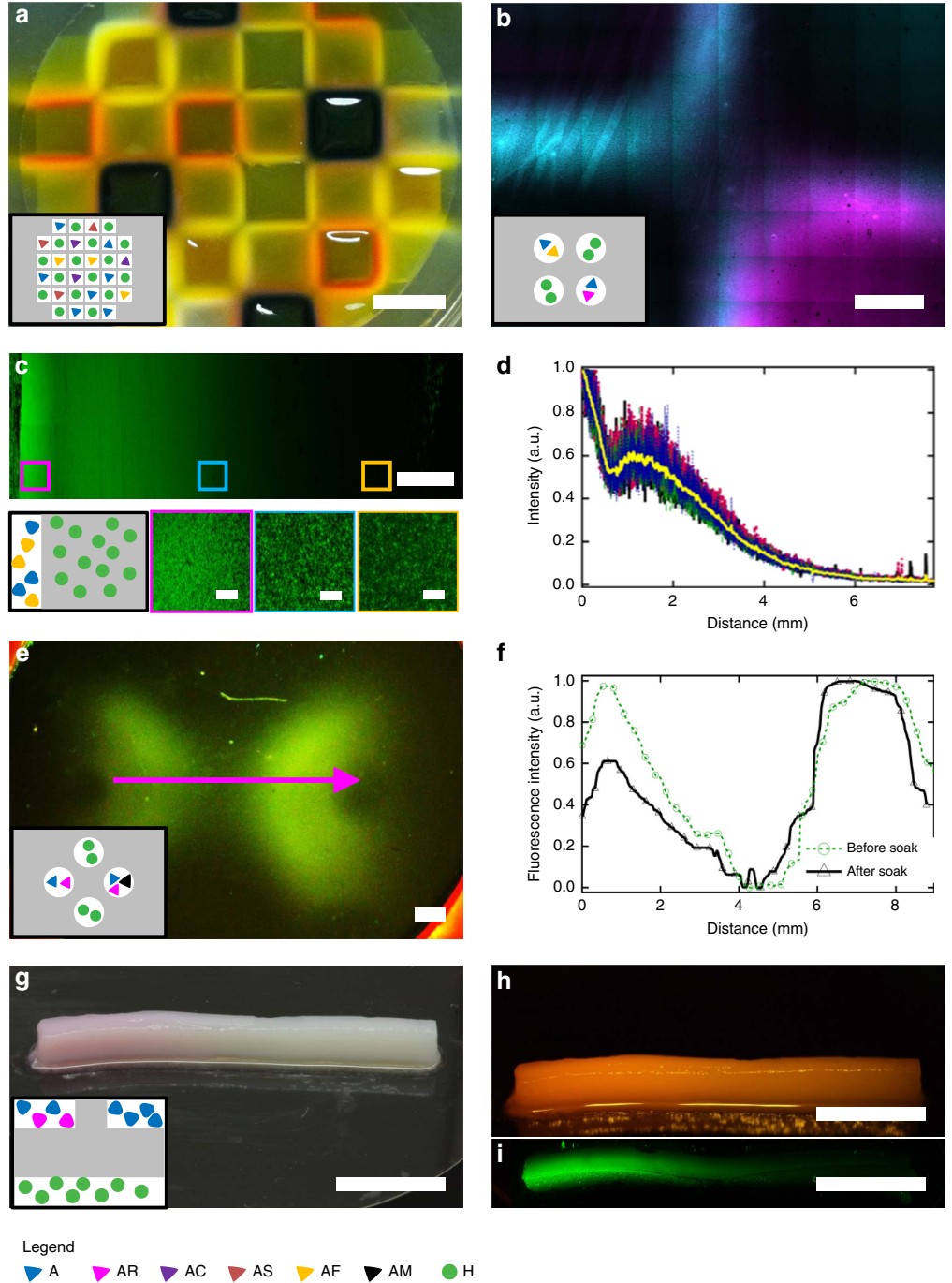

**Figure 5 | Functional and gradient patterns and objects obtained by RD-SA.** (**a**) A large grid made with several benzaldehyde-functionalized dyes, (yellow, orange, and purple), added for chemical differentiation within a single structure (Supplementary Methods). (**b**) Chemically differentiated gel objects, with two aldehyde-functionalized dyes (**AF** with $\lambda_{ex} = 488$ nm and **AR** with $\lambda_{ex} = 543$ nm) incorporated into the gel fibres, as imaged by confocal fluorescence microscopy. The image is composed of $10 \times 8$ individual micrographs, leading to visible edges (Supplementary Methods). (**c**) A permanent chemical gradient formed by letting **A** diffuse from the left into an agar gel containing **H**, imaged by fluorescence imaging (Supplementary Methods). The confocal micrographs below show magnifications of the gradient, highlighting the change in fibre density. (**d**) Fluorescence intensity measured from left to right in **c**. The yellow line is the average of multiple lines in the same image. (**e**) Fluorescence image of **ConA** bound to a gel pattern made by RD-SA, in which mannose groups are incorporated in the structure on the right. The image was recorded after partially removing unbound **ConA** by soaking the structure in buffer (Supplementary Methods). (**f**) Fluorescence intensity measured in **e**, along the magenta arrow. The green dashed data is before soaking in buffer, black is after soaking in buffer. (**g**) Photograph of a free-standing bar of alginate/**HA₃** with incorporated gradient of fluorescence (**AR** was used as a fluorophore). (**h**) Photograph of the object in **g** illuminated with an LED (540 nm). The photo was taken through a filter (cutoff wavelength was 580 nm). (**i**) A confocal micrograph of the object in **g**. Scale bars: 1 cm in **a** and **g**–**i**; 1 mm in **b**, **c** (top) and **e**; 50 μm in **c** (bottom). Schematic insets in **a**, **b**, **c**, **e** and **g** show the scheme of the formation of corresponding patterns. The grey areas denote the gel matrix and the white areas denote the reservoirs. **H** is hydrazide; **A** is aldehyde; **AC** is benzaldehyde labelled with cyanine; **AF** is benzaldehyde labelled with fluorescein; **AP** is benzaldehyde labelled with sulfonated pyrene; **AR** is benzaldehyde labelled with rhodamine B; **AS** is benzaldehyde labelled with styryl; **AM** is benzaldehyde labelled with mannose.

the formation of a band as the normalized fluorescence had a local minimum at ~1 mm distance from the source of **A**, after which it gradually decreases towards the end. This phenomenon was observed in repeated experiments and shows up in both local fibre density as well as in the summed fluorescence intensity, and may have an origin similar to the Liesegang patterns observed in precipitation systems[7,34]. Although in principle this phenomenon could have occured in all of our other experiments, we observed it only in the experiments when pH was ~7.0 (Supplementary Fig. 12).

To further capitalize on the potential of chemical differentiation of organic materials made by RD-SA, we attempted to functionalize these structures with molecular recognition sites for proteins. Here, the modular nature of the self-assembling system employed in this work plays a pivotal role[32]. We used the extensively described non-covalent binding of the lectin Concanavalin A (**ConA**) to mannose as a protein-ligand interaction[35]. Using the mannose-functionalized benzaldehyde **AM**, we loaded the reservoirs of a Plexiglass holder with **A** and **A** + **AM**, and allowed diffusion and reaction with **H** using the RD-SA approach shown in Supplementary Fig. 2. This resulted in the formation of millimetre-scale supramolecular shapes within the agar matrix. To test **ConA** binding, we loaded the holder such that one of the two formed shapes was labelled with mannose. Next, fluorescein-labelled **ConA** was allowed to diffuse into the matrix, to bind to the mannose groups on the fibres. The entire matrix was subsequently soaked in buffer solution for several days to remove unbound or non-specifically bound protein. Fluorescence microscopy shows that, after soaking, the amount of **ConA** decreases more on the fibre structures without mannose functional groups, when compared to those that do contain mannose (Fig. 5e,f). The relative stability of **ConA** on the mannose-functionalized supramolecular structures shows the potential of RD-SA to chemically differentiate supramolecular structures with biological functionalities in space.

Finally, we combined several demonstrated principles to make a free-standing object with a permanent chemical gradient. To achieve that, we positioned solutions of **H**, **A**, and **A** + **AR** (rhodamine B-functionalized benzaldehyde) into reservoirs in an alginate matrix. After 24 h, we dissolved the alginate and the formed object was left standing in a large amount of water overnight to remove unreacted compounds, after which it was imaged. As can be seen in Fig. 5g, the red colour intensity gradually decreases along the object from left to right, showing a gradient of functionalization along the object. The same gradient was also visualized using fluorescence (Fig. 5h,i).

**Quantitative analysis of 1D pattern formation**. To better understand pattern formation through RD-SA, we developed a simple RD model describing the formation of the line structure in the basic experiment shown in Fig. 2a. The required reaction rate constants were determined using a kinetic model in which we considered the **HA₃** formation reaction as a 3-step forward reaction (Supplementary Figs 7 and 8, Supplementary Table 2, and Supplementary Methods), followed by a gelation step. Diffusion of species was described using Fick's first law, where we set the diffusion coefficient of **HA₃** close to zero to incorporate the gelation step. We solved the RD model (Supplementary Methods) for a range of diffusion coefficients for the other species and obtained their value (see Supplementary Table 3) by finding the best fit between simulated and experimentally observed temporal development of the width of the 1D pattern at pH = 4.0 (see Supplementary Fig. 6 for details about the determination of width). We note that it is important to hereby take into account that the diffusion depends on the local formation of **HA₃** gel,

which was done by considering diffusion coefficients that depend on the local concentration of **HA₃** (Supplementary Fig. 9 and Supplementary Methods). Figure 6a shows the resulting spatial and temporal variation of the concentrations of **H** (green), **A** (blue), and **HA₃** (yellow) along the gel. It can be seen that the temporal concentration profile of **HA₃** from the model resembles the measured temporal intensity profile of line formation as shown in Fig. 6b. Since change in intensity is related to the formation of **HA₃**, this result is in good qualitative agreement with the experimental data. Next, we used the model to predict the response of the system to RD parameters that can be easily controlled in an experiment—namely, the diffusion distance, the initial reactant concentrations, and pH (controlling the reaction rates). The model predicts, after reaching a steady value after about 50 h, a weak dependence of the width of the 1D pattern on the diffusion distance (Supplementary Fig. 10), while a much stronger dependence was predicted for the pH. According to the model, reducing the pH from 7 to 3.3 causes a fivefold decrease of the line width, which is in excellent quantitative agreement with the experimentally observed line widths within this pH range (Fig. 6c). Because the pH mainly influences the reaction rates (see Supplementary Figs 4 and 5, Supplementary Table 1 and Supplementary Methods for details), these results clearly indicate that within the current experimental set-up the reaction rates for hydrazone formation limit the minimum attainable width to ~2 mm (see Supplementary Discussion for extended explanation). The fabrication of objects with smaller structural features would require either substantially higher reaction rates and hence different chemistry, and/or a different experimental set-up. Here, the WETS approach, although with its own limitations, has been shown to be an excellent tool for further downscaling of the pattern dimensions.

In conclusion, we have demonstrated that SA of a supramolecular gelator can be coupled to RD to fabricate free-standing objects of variable size, shape, and chemical functionality. The objects can vary in size from several hundred micrometres to centimeters, and chemical functionalities such as fluorophores or molecular recognition sites can be easily incorporated in this modular system. Through RD, there is control over the location and density profile of chemical functionality in these gel objects. Taking into account the versatility of self-assembling systems, and the vast number of organic reactions, this approach could be easily extended to any organic system for production of functional materials with defined shapes, sizes, and functionalities.

## Methods
**Materials**. All reagents were purchased from commercial sources and used as provided, unless stated otherwise. Hydrazide **H**, aldehyde **A**, aldehydes labelled with cyanine (**AC**), fluorescein (**AF**), rhodamine (**AR**), styryl (**AS**), and mannose (**AM**) were synthesized according to reported methods[27,32]. All experiments were performed using MilliQ water as a solvent, unless mentioned otherwise. All stock solutions of **A**, **H**, or labelled aldehydes were prepared in 100 mM phosphate buffer (pH = 4.0). The stability of the pH was checked by measuring the pH before and after the experiment and no significant differences were detected.

Press-to-Seal silicone isolators with adhesive, one well, were bought from Life Technologies. They were 20 mm in diameter and 0.5 mm deep.

**Equipment**. Ultraviolet–visible spectroscopic measurements were performed on an Analytik Jena Specord 250 spectrophotometer. Confocal Laser Scanning Microscopy micrographs were obtained using a Zeiss LSM 710 confocal laser scanning microscope, equipped with ×2.5 and ×10 air objectives and a ×40 oil immersion objective. The sensitivity of detectors and filters was adjusted to obtain maximum signal to noise ratio. Time-lapse movies were recorded with a MicroCapture USB camera connected to a standard desktop computer, with suitable software. Data analysis were performed using MATLAB 2012b and ImageJ (http://imagej.nih.gov). The compression test were performed using a dynamic mechanical analyzer DMA 7e (Perkin Elmer Instruments).

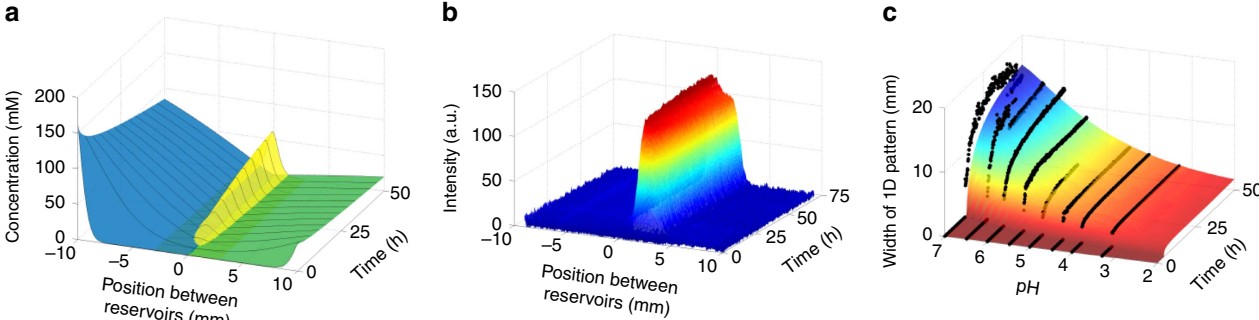

**Figure 6 | Predicting control over 1D pattern dimensions using RD modelling. (a)** The temporal concentration profiles of hydrazide **H** (green), aldehyde **A** (blue) and gelator **HA₃** (yellow) obtained from the model (pH = 4.0). (**b**) The temporal intensity profile along the distance between the reservoirs of **H** and **A**, as obtained from experiment (pH = 4.0). The increase in intensity signifies the formation of **HA₃**. (**c**) Comparison between the RD model (coloured surface) and experiment (black dots). The reaction rate in the experiments was controlled by varying the pH between 3.3 (fast reaction) and 7.0 (slow reaction).

**Preparation of agar gels.** Phosphate buffer was prepared at the desired pH by dissolving the appropriate amounts of acid–base pair of phosphate salts at a total concentration of 100 mM in water. If necessary, the pH was adjusted with NaOH or $H_3PO_4$ solutions. A diffusion matrix was made of 1% agar (mass fraction, $w$) by dissolving agar in the appropriate amount of phosphate buffer. The experiments were conducted using this composition of diffusion matrix, unless stated differently. To completely dissolve the agar, the mixture was heated and stirred at ~100 °C until the solution turned transparent. The solution was subsequently allowed to cool down and left to solidify.

**Preparation of alginate gels.** Alginate gels were prepared according to the procedure proposed by Draget et al.[36] Briefly, a powder of $CaCO_3$ was dispersed in water. Then, dry sodium alginate was added and this dispersion was heated at 100 °C in a closed vial until all alginate and $CaCO_3$ was dissolved. The solution was subsequently allowed to cool to room temperature. Finally, an aqueous solution of glucono-$\delta$-lactone (**GDL**) was added, and the resulting solution was stirred briefly and transferred to a Petri dish. 10 ml of solution per Petri dish was used. The alginate gel was formed by leaving the solution standing overnight. Unless otherwise stated, final concentrations of $CaCO_3$, alginate, and **GDL** were 12.5 mM, 1.5% (mass fraction, $w$), and 40 mM, respectively, giving alginate gels with pH = 4.5. To prepare alginate gels labelled with BODIPY TR, 3% (mass fraction, $w$) of alginate-BODIPY TR was added with respect to the total mass of alginate.

**RD-SA experiments by cutting reservoirs in agar gels.** To prepare the agar gel matrix for RD-SA experiments, 10 ml of agar solution was heated until dissolved. When still warm, this solution was poured in a plastic Petri dish (diameter ca. 50 mm, height ca. 15 mm) and left to gelate upon cooling. The same type of Petri dishes were used in other experiments, unless stated otherwise. A slab of the agar matrix with desired dimensions was made by manually cutting two parallel lines at equal distance from the centre of the Petri dish for the experiment in Fig. 2a. The two outer segments of agar were removed, forming two reservoirs for the solutions of **H** and **A**. These reservoirs were completely separated from each other, as the agar gel extends fully to the sides of the Petri dish, ensuring that no mixing of solutions can occur other than by diffusion through the gel matrix. Solutions of **H** (40 mM) and **A** (160 mM) were prepared in the same buffer as used for the agar gel preparation. The two solutions were then pipetted in the two reservoirs and left to diffuse in a closed humidified environment, which prevented evaporation of liquids. The experiment in Fig. 2b was performed using a similar approach, except for making reservoirs in the agar with non-straight edges, and for filling the reservoirs with solutions of **A** and **H** in agar instead of in buffer. For the experiment with the two-dimensional grid of reservoirs in Fig. 2f, reservoirs with a diameter of 5 mm were cut in the gelated agar and filled with buffer solutions of **A** and **H** in an alternating pattern.

**RD-SA by using a PDMS mould.** To form RD-SA patterns in a flat agar matrix without reservoirs at the sides, we used a two layer approach. The first layer is a PDMS mould with reservoirs for H and A, and the second layer is a flat agar matrix in which the RD-SA patterns form. The PDMS mould was either made by manually cutting a layer of PDMS cured in a Petri dish in the form of the circular and triangular mould shown in grey in Fig. 2c,d, or by curing liquid PDMS in a Petri dish against a Teflon mould containing an array of squares (8 × 8 × 3 mm, 2 mm separation between them; Fig. 2e). The reservoirs were completely filled with agar solutions containing H (40 mM) and A (160 mM). The PDMS mould with the agar was left to cool down to room temperature, to allow the agar to gelate. Then, a new agar layer was poured on top of the PDMS mould, which serves as the diffusion and reaction layer in which the RD-SA patterns form. After gelation

of this layer, the Petri dish was placed in a closed humidified environment. After 24–78 h it was taken out and the PDMS mould was gently removed from the agar layer by pushing a glass slide between the PDMS mould and the agar layer, which contained the formed RD-SA patterns.

**The formation of free-standing macroscopic hydrogel objects.** The experiments were conducted in Petri dishes. Alginate gels were prepared according to the described procedure. Reservoirs were manually cut into the gel in various configurations shown in Fig. 4b. The solutions of **H** (40 mM) and **A** (160 mM), both prepared in phosphate buffer of pH = 4.5, were placed in the reservoirs. Depending on the configuration, the reservoirs can hold up to 2.5 ml of solution. After the sample was left overnight, the remains of the solutions of **H** and **A** were removed. Free-standing patterns were obtained by adding a solution of EDTA (0.5 M) into the Petri dish and waiting until the alginate was dissolved (confirmed by visual inspection). Then, the resulting solution was removed with a pipette and the remaining hydrogel pattern was washed several times with water.

The free-standing gradient object shown in Fig. 5g–i was made following a similar procedure with the arrangement shown in Supplementary Fig. 2e.

**RD by wet stamping.** Three percentage (mass fraction, $w$) agar stamps were used in all WETS experiments. Stamps were prepared by dissolving agar in an appropriate amount of water (heated to 100 °C) and casting the hot solution on PDMS moulds in a Petri dish. After cooling, gel stamps were cut out and soaked in a solution of **H** (40 mM) + **AR** (30 μM ) for at least 12 h. Gels used as substrates were 2% (weight/volume) alginate. The substrates were prepared by dissolving alginate in an appropriate amount of water (heated to 100 °C) and casting the cold solution into the well made by sticking two Press-to-Seal silicone isolators on top of each other on a glass slide (giving a height of 1 mm). A dialysis membrane was placed on top and gently pressed. The membrane was used to allow diffusion of calcium into the liquid alginate and to prevent alginate from leaving the well. The whole set-up was then immersed into a 100 mM solution of $CaCl_2$ for at least 12 h. Subsequently, the membrane was gently removed and the substrate was soaked in a solution of **A** (300 mM) for at least 12 h. The surfaces of the substrate and the stamp were air-dried before stamping. The stamp was gently put on the substrate and left for 60 min, after which the stamp was removed. The substrate was left standing overnight in a humid atmosphere to prevent drying. Alginate substrates were dissolved with a 0.5 M solution of EDTA by pipetting several drops of the solution onto the substrate. When all alginate was dissolved (confirmed by visual inspection), the solution was removed by careful pipetting and the remaining objects were carefully washed with water. Imaging of substrates and free-standing objects was done with a MicroCapture USB microscope camera and a confocal fluorescence laser scanning microscope. The solutions of **H** (40 mM) + **AR** (30 μM) and **A** (300 mM) were prepared in phosphate buffer of pH = 4.5.

**Data availability.** The data that support the findings of this study are available from the corresponding authors upon reasonable request.

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

## Acknowledgements

We thank J.M. Poolman, L. van de Mee, F. Trausel and F. Versluis for providing compounds **H** and **A**. Further, we thank A. Olive and J.M. Besselink for measuring the rate constants of the model reaction. Furthermore, we thank B. Norder for help with compression tests. This work is supported by the Dutch Organisation for Scientific Research (Vidi grant to R.E., Complexity programme R.E., J.H.v.E. and W.E.J.H.), Marie Curie initial training network 'SMARTNET' (project no. 316656; J.H.v.E. and M.L.) and NanoNextNL, a micro and nanotechnology consortium of the Government of the Netherlands and 130 partners (project no. 07 A.11; J.H.v.E. and M.L.).

## Author contributions

M.L., W.E.J.H., R.E. and J.H.v.E. designed experiments; M.L. performed experiments and modelling; W.E.J.H. performed **ConA** binding and 'writing' our research group's name; C.M. synthesized compounds **AC**, **AF**, **AR**, **AS** and **AM**; S.M. synthesized **AP** and helped with the labelling of alginate; V.v.S. analysed the modelling data and helped with the development of the model; M.L., R.E. and J.H.v.E. analysed data and wrote the manuscript; R.E. and J.H.v.E. supervised the project. All authors edited and approved the final manuscript.

## Additional information

**Competing interests:** The authors declare no competing financial interests.

DOI: 10.1038/ncomms16128    OPEN

# Erratum: Free-standing supramolecular hydrogel objects by reaction-diffusion

Matija Lovrak, Wouter E.J. Hendriksen, Chandan Maity, Serhii Mytnyk, Volkert van Steijn, Rienk Eelkema & Jan H. van Esch

Nature Communications 8:15137 doi: 10.1038/ncomms15317 (2017); Published 5 Jun 2017; Updated 30 Jun 2017

In the original HTML version of this Article, which was published on 5 June 2017, the publication date was incorrectly given as 5 July 2017. This has now been corrected in the HTML; the PDF version of the paper was correct from the time of publication.

