## [Peer Review File · Nature Communications]

Editorial Note: Parts of this peer review file have been redacted as we could not obtain permission to publish the reports of reviewer #2.

Reviewers' comments:

Reviewer #1 (Remarks to the Author):

This paper describes the production of freestanding supramolecular hydrazone-based hydrogels with a range of shapes and sizes as well as the possibility of building in gradients and chemical functionalities by varying the nature and initial spatial distribution of the reactants. The experimental work is supported by modeling studies. The supporting information provides considerable detail that should make it possible for other laboratories to reproduce the work.

The paper is very well written and organized. The work appears to have been carefully done. This is essentially a proof of principle that demonstrates how one might construct freestanding gel objects by utilizing a substrate that can be removed by chemical means once reaction-diffusion processes initiated from a well-chosen set of starting conditions have created the desired structure. As the authors note, this approach offers considerable promise for constructing objects and devices with a powerful variety of functions.

I would raise a few questions for the authors to consider in any revised version.

1) As the authors point out, the phenomenon bears a resemblance to Liesegang ring formation. They mention band formation on p. 12 in connection with their gradient experiments, but it is not clear, at least to this reader, why band formation does not occur in the experiments described earlier. Is it simply that formation of the RD structure blocks diffusion through the gel? If so, perhaps this could be stated explicitly at an early stage and the key differences between these experiments and Liesegang-type experiments made clear.

2) Is it possible to obtain order of magnitude estimates of the maximum and/or minimum sizes of objects that can be produced by this technique from the rate and diffusion coefficients in the model?

3) In the description of the modeling in the supporting information, it would be helpful to specify the value of k_2 above which the results are independent of k_2 rather than simply saying $k_2 \gg k_1, k_3$.

In summary, this is a very nice piece of work that certainly merits publication. There appears to be a considerable literature on using reaction-diffusion to generate structures, with which I am not totally familiar. If the advances described here - free standing gels and control over size and functionality - are truly novel, then I am happy to recommend publication in Nature Communications.

Reviewer #2 (Remarks to the Author):

[redacted]

Reviewer #3 (Remarks to the Author):

In this paper, Lovrak and co-workers describe the reaction-diffusion self-assembly (RD-SA) strategy to

fabricate free-standing polymer and supramolecular composite hydrogel. They achieved formation of spatially complex shapes and patterns (waves, grid, and so on). Also, the composite of alginate and supramolecular nanofibers shows a largely increased yield stress. They successfully demonstrated that the RD-SA method is quite useful for spatially-patterned and/or gradient presentation of functional organic and biomolecule (protein), leading to functional materials possessing different functionalities and (bio)molecules in a spatially-defined manner.

As they mentioned, there are examples using the reaction-diffusion strategy for not only inorganic but also organic and biomolecules. However, their demonstration for the self-assembled hydrogelators has high novelty for publication, and it is expected that their RD-SA method can be widely applied to other self-assembled system. Yet, this reviewer considers that characterization of the resulting self-assembled aggregates is not sufficient in this stage, thus the presented data still leaves open questions.

Comments:

1. On Figure 3, page 8, they showed confocal images of fluorescently stained HA3 and alginate. I'm very interested in and have several questions/comments about this mixed structure as shown below.

(i) Does this HA3/alginate mixed structure remain after treatment of EDTA?

(ii) I'm interested whether the supramolecular structure of HA3 in alginate gel is the same as in buffer (without any polymer gels). Are there any optical measurements to check the self-assembled structures, such as CD spectroscopy?

(iii) In this paper, the authors mentioned this mixed structure as "double network gels." According to confocal images, however, the two networks seem well merged. Thus, I do not think that the word, "double network", is appropriate. Please use other proper words instead of "double network." Or, please show any evidence to prove whether the structures of HA3 and alginate formed into double-networks each other?

(iv) Kiriya et al. already discussed the wrapping structure of supramolecular nanofibers by alginate gels (Angew. Chem. Int. Ed. 2012, 51, 1553 –1557). This paper may help the authors to discuss the mixed structure of alginate and supramolecular nanofibers.

2. Throughout the article, the authors use 1% agar or 1.5% alginate gels as the diffusion matrix. What will happen if the concentration of agar and alginate decreases? Is there a critical concentration to form the RD pattern of supramolecular gels by increasing diffusion coefficients of precursors?

3. The authors used phosphate buffer (pH 4.0) for making agar gel. However, it is clear that the buffering region of phosphate does not cover around pH 4.0. Is there any special reason for using phosphate buffer? If so, please describe it somewhere.

Reviewer #1 (Remarks to the Author)

I would raise a few questions for the authors to consider in any revised version.

1) *As the authors point out, the phenomenon bears a resemblance to Liesegang ring formation. They mention band formation on p. 12 in connection with their gradient experiments, but it is not clear, at*

least to this reader, why band formation does not occur in the experiments described earlier. Is it simply that formation of the RD structure blocks diffusion through the gel? If so, perhaps this could be stated explicitly at an early stage and the key differences between these experiments and Liesegang-type experiments made clear.

The main difference between the “gradient experiment” and the “other reported experiments” is the pH, which strongly influences the time scale for reaction relative to the time scale for diffusion. We agree with the mechanism described by the referee; at low pH, the fast formation of the RD structure blocks diffusion through the gel such that Liesegang patterns will not form. We indeed observed Liesegang patterns in experiments other than the “gradient experiment”, when performing them at higher pH (close to 7 instead of 4 as in Figure 2).

We clarified this point by (1) adding the following sentence to page 11 of the manuscript: “Although in principle this phenomenon could have occurred in all of our other experiments, we observed it only in the experiments when pH was around 7.0 (Supplementary Fig. 14).”, and (2) adding a Supplementary Fig. 14 to the SI showing Liesegang patterns for pH = 6.5 and 7.0.

2) Is it possible to obtain order of magnitude estimates of the maximum and/or minimum sizes of objects that can be produced by this technique from the rate and diffusion coefficients in the model?

Yes it is. We clarified this point by adding the following explanation at the end of Supplementary Discussion in Supplementary Information (page 21): “Additionally, by knowing only the reaction rates and the diffusion coefficients of participating species, the model can be used to estimate minimum and maximum sizes of produced patterns and is not limited to the reaction used in this research. In our case, the minimum size estimated using the model was in the order of 2 mm, and maximum size was in the order of 15 mm, thus spanning over approximately one order of magnitude.”

3) In the description of the modeling in the supporting information, it would be helpful to specify the value of k_2 above which the results are independent of k_2 rather than simply saying $k_2 \gg k_1, k_3$.

The value of k_2 has to be at least $1000 \times$ higher than k_1 and k_3 . Then, the concentration of intermediate **HA** is negligible at all stages of **HA₃** formation which is in agreement with the experimental results shown in Supplementary Fig. 10.

We clarified this point by adding the following explanation to page 18 of the Supplementary Information: “ k_2 was at least $1000 \times$ higher than k_1, k_3 ”.

Reviewer #2 (Remarks to the Author)

[redacted]

Reviewer #3 (Remarks to the Author)

1. On Figure 3, page 8, they showed confocal images of fluorescently stained HA3 and alginate. I'm very interested in and have several questions/comments about this mixed structure as shown below.

(i) Does this HA3/alginate mixed structure remain after treatment of EDTA?

We have not investigated the structures after treatment with EDTA using confocal microscopy. However, we investigated the mechanical properties of alginate/HA₃ after treatment with EDTA. It can be seen in figure below that the yield stress of the alginate/HA₃ hybrid network decreases upon treatment with EDTA. Although this decrease might appear substantial, the remaining network is still 10 times stronger than pure 30 mM HA₃ (683 ± 187 Pa). This result suggests that, although some alginate might have been removed, the alginate/HA₃ hybrid network structure largely remains intact.

(ii) I'm interested whether the supramolecular structure of HA3 in alginate gel is the same as in buffer (without any polymer gels). Are there any optical measurements to check the self-assembled structures, such as CD spectroscopy?

We thank the referee for the valuable suggestion to use CD to investigate the supramolecular structure of the system. We attempted this but unfortunately all measurements failed because of extensive scattering of the samples. Instead, we performed IR measurements on the powder of HA₃ and on the powder of HA₃/agar structures. Both spectra showed amide I (1655 cm⁻¹) and amide II (1513 cm⁻¹) peaks which are indications of hydrogen bonding, thus suggesting that the structure of HA₃ aggregates in the alginate environment and in buffer are comparable.

(iii) In this paper, the authors mentioned this mixed structure as "double network gels." According to confocal images, however, the two networks seem well merged. Thus, I do not think that the word, "double network", is appropriate. Please use other proper words instead of "double network." Or, please show any evidence to prove whether the structures of HA3 and alginate formed into double-networks each other?

We agree with the referee that “double network gels” might not be appropriate term to describe alginate/HA₃ network. Since we used confocal microscopy for imaging alginate/HA₃ network, we cannot unambiguously distinguish how exactly two networks are assembled due to the spatial resolution of confocal microscope.

We addressed this concern by replacing the term “double network” with the term “hybrid network” wherever applicable.

(iv) Kiriya *et al.* already discussed the wrapping structure of supramolecular nanofibers by alginate gels (*Angew. Chem. Int. Ed.* 2012, 51, 1553 –1557). This paper may help the authors to discuss the mixed structure of alginate and supramolecular nanofibers.

Indeed, the suggested paper helped us to strengthen our hypothesis.

We addressed this point by adding the following clarification on page 8 of the manuscript, which includes the suggested reference as reference 30: “We did not investigate how the interactions or synergy between HA₃ and alginate lead to improved mechanical properties, but we hypothesize that HA₃ fibers wrap around the alginate chains and create cross-links between the alginate chains in addition to already existing calcium cross-links, most likely in a similar fashion as recently has been reported by Kiriya *et al.*³⁰ This additional crosslinking, in turn, would improve the mechanical properties of alginate/HA₃ hybrid network material.”

2. Throughout the article, the authors use 1% agar or 1.5% alginate gels as the diffusion matrix. What will happen if the concentration of agar and alginate decreases? Is there a critical concentration to form the RD pattern of supramolecular gels by increasing diffusion coefficients of precursors?

Solutes typically diffuse 30% slower in 1% agar than in pure water and are about 80% slower in 1.5% alginate than in pure water (Amsden, *Macromolecules*, **31**, 1998). Therefore, if the concentrations of agar and alginate decrease, the diffusion coefficients of H and A would increase. This increase would lead to wider patterns according to our model. In fact, we even managed to prepare HA₃ patterns in buffer solutions (without agar or alginate). This finding suggests that there is no critical concentration of agar or alginate to form RD patterns. However, in the gel-free buffer media, the patterns are typically severely distorted by convective mixing of the reactants.

3. The authors used phosphate buffer (pH 4.0) for making agar gel. However, it is clear that the buffering region of phosphate does not cover around pH 4.0. Is there any special reason for using phosphate buffer? If so, please describe it somewhere.

Phosphate has $pK_1 = 2.12$. Since optimal buffer capacity is achieved in the region $pK \pm 1$, phosphate buffer at pH = 4 is not optimal but still somewhat close to pK_1 . In our previous work on the HA₃ system we used phosphate buffer as a medium for gelation (ref 27 and 28 in the manuscript). To avoid any possible effects of different ions on gelation, we decided to keep ion composition the same. To check that the pH of stock solutions was maintained, we always measured pH before doing experiments and adjusted it if necessary. We also observed that pH during RD experiments remained unchanged.

We clarified this point on page 16 of the manuscript (Methods section) by adding the following sentence: “The stability of pH was checked by measuring pH before and after the experiment and no significant differences were detected.”

Other changes

In addition to the changes as indicated above, we have also moved most of the methods description from the Supporting Information to the main manuscript and have reorganized the Supporting Information to comply with the editorial guidelines. We have also corrected a few grammatical and spelling errors. All changes to the manuscript are marked.

REVIEWERS' COMMENTS:

Reviewer #1 (Remarks to the Author):

The authors have responded satisfactorily to the issues raised in my original report. I recommend that the article be accepted.

Reviewer #3 (Remarks to the Author):

I went through the revised manuscript. I think that the authors properly replied to my comments and it is now acceptable.